# Current Overview of Spinocerebellar Ataxia Type 7 in Mexican Population: Challenges in Specialized Care for a Rare Disease

**DOI:** 10.3390/ijms251910750

**Published:** 2024-10-06

**Authors:** César M. Cerecedo-Zapata, Yessica S. Tapia-Guerrero, José A. Ramírez-González, Aranza Meza-Dorantes, Karla N. Tercero-Pérez, Hernán Cortés, Araceli Guerra-Grajeda, Ilse H. Ortega-Ibarra, Gabriela Gatica-Ramos, Alfredo Poblete-Velazquez, Norberto Leyva-García, Luis Velázquez-Pérez, Bulmaro Cisneros, Jonathan J. Magaña

**Affiliations:** 1Laboratory of Genomic Medicine, Department of Genetics, National Rehabilitation Institute Luis Guillermo Ibarra Ibarra (INRLGII), Mexico City 14389, Mexico; misael_207@hotmail.com (C.M.C.-Z.); yessicasarai@gmail.com (Y.S.T.-G.); jalfredormz88@gmail.com (J.A.R.-G.); hcortes_c@hotmail.com (H.C.); grajeda.ara@gmail.com (A.G.-G.); nleyva@inr.gob.mx (N.L.-G.); 2Rehabilitation and Special Education Center of Veracruz (CRISVER), Xalapa de Enriquez 91097, Mexico; k.tercero.p@gmail.com; 3Department of Bioengineering, School of Engineering, Tecnologico de Monterrey, Campus Ciudad de México, Mexico City 14380, Mexico; a01730904@tec.mx; 4Instituto de Salud Pública, Universidad Veracruzana, Xalapa de Enriquez 91000, Mexico; ihoi@bizendaa.unistmo.edu.mx (I.H.O.-I.); gabyzzi10@gmail.com (G.G.-R.); 5Hospital Regional de Xalapa “Dr. Luis F. Nachón”, Secretaria de Salud de Veracruz, Xalapa de Enriquez 91130, Mexico; pobletevelazquezmedico@gmail.com; 6Science Cuban Academy, La Havana 12400, Cuba; velazq63@gmail.com; 7Department of Genetics and Molecular Biology, Center of Research and Advanced Studies (CINVESTAV-IPN), Mexico City 07360, Mexico

**Keywords:** spinocerebellar ataxia type 7, prevalence, genetic testing, CAG repeats, Mexican population

## Abstract

Spinocerebellar ataxia type 7 (SCA7) is a rare genetic disease characterized by progressive cerebellar syndrome and macular degeneration. In a previous study, we clinically and genetically characterized a group of Mexican patients, which represented one of the largest cohorts of SCA7 patients worldwide and demonstrated that all patients had a unique genetic origin. Our laboratory developed a program for the diagnosis, medical care, and long-term follow-up of these patients living in Veracruz State, and in this report, we present an update to this research, covering 2013 to 2024. So far, we identified 172 SCA7 carriers, with a few cases outside Veracruz, and our data support that the length of the CAG repeat tract mainly determines disease severity and life expectancy, and accordingly, we define three different phenotypes, early-onset (EO), classical-onset (CO), and late-onset (LO), with EO patients showing the lowest life expectancy. Furthermore, we found that parental transmission of mutant alleles leads to increased CAG repeat instability, compared to maternal ones. Interestingly, a haplotype analysis revealed that patients outside Veracruz may have different genetic origins. In conclusion, longitudinal observations of SCA7 patients provide insight into the natural history of SCA7 and help to design strategies for diagnosis, genetic counseling, physical rehabilitation, and therapeutic alternatives.

## 1. Introduction

Spinocerebellar ataxia type 7 (SCA7) is an autosomal dominant neurodegenerative disease caused by the pathological expansion of a Cytosine-Adenine-Guanine (CAG) trinucleotide repeat located in the *ATXN7* gene [1]. SCA7 presents a significant motor and visual disability, characterized by different clinical manifestations, including cerebellar and phoniatric dysfunction, pigmentary retinal degeneration, gait ataxia, dysdiadochokinesia, dysmetria, dysarthria, sensory loss, hyperreflexia, and postural tremor [2]. The size of CAG repeats ranges from 4 to 18 in normal alleles and from 36 to 460 repeats in expanded alleles [1]. It is known that the length of the CAG tract correlates directly with the severity of the disease and inversely with age at onset [3,4,5]. In this regard, two main SCA7 phenotypes have been described: early-onset (EO) and adult-onset (AO) [6]. The EO phenotype is characterized by full penetrance in patients with ≥47 CAG repeats and is associated with a significant motor and visual clinical picture. In contrast, the AO phenotype typically manifests in patients with alleles ranging from 36 to 46 repeats and is generally characterized by autonomic disturbances, sleep disorders, and extrapyramidal and movement symptoms [6]. Nevertheless, the lack of a sufficiently large cohort of patients has precluded the possibility of conducting detailed clinical characterizations.

The disease has been identified in families from various ethnic backgrounds in multiple countries worldwide, including France, Belgium, Germany, the United Kingdom, Algeria, Morocco, Libya, Tunisia, Zambia, South Africa, Israel, South Korea, the Philippines, Australia, Brazil, the USA, and Mexico, among others [5,7,8,9,10,11,12]. Although relatively high prevalence rates of SCA7 have been described in countries such as Sweden, South Africa, and Mexico, as a consequence of founder effects, it is a rare disorder that barely reaches a global prevalence of <1/100,000 [13,14,15,16,17]. Isolated cases have even been reported in specific populations in Latin American countries such as Mexico, Brazil, and Cuba, leading to a lack of accurate data on the prevalence of this pathology at the national level [18,19]. As a result, SCA7 is considered an orphan or rare disease, and national health systems often lack a systematic methodology for managing these patients. Therefore, early diagnosis, the knowledge of prevalence, the identification of presymptomatic individuals, and large-scale studies to describe the clinical features of patients and the course of the disease are essential to design management strategies.

In this context, it is important to highlight the discovery of a significant cluster of SCA7 cases within a relatively small area of 1200 km^2^ in the southeastern region of Mexico (Veracruz State). This area is surrounded by mountain ranges [11], which have hindered communication with other populations and restricted access to clinical services. The discovery of this population has allowed us to conduct extensive research on SCA7 in clinical and scientific areas, which has increased the knowledge of this pathology [6,7,11,20,21,22,23,24,25,26]. Therefore, the present study aims to describe the current status and new findings of this key SCA7 population, including clinical, genetic, demographic, and epidemiological aspects, highlighting their contribution to the diagnosis, clinical management, and potential treatment of SCA7. We believe that monitoring this population has provided a better outlook on the overall SCA7 disability and a deeper comprehension of the natural history of the disease, thereby enabling the design of long-term trials.

## 2. Results

### 2.1. Establishment of Molecular Diagnosis and Epidemiologic Data

The initial exploration of the central region of Veracruz was conducted with the objective of implementing a clinical research protocol for SCA7, including identifying subjects with a symptomatology of SCA7 (index cases) through clinical evaluation and molecular diagnosis, and the identification of relatives at risk of carrying a pathological allele by the construction of family pedigrees, as SCA7 is an autosomal dominant disease (Figure 1). From July 2013 to July 2024, 192 individuals at risk were enrolled in the clinical research protocol, of whom 80 were identified as SCA7-positive through genetic testing (Table 1). It is noteworthy that 30 of these carriers did not present any symptoms, whereas among the 50 symptomatic patients, 36 exhibited the AO phenotype and the remaining 14 exhibited the EO phenotype. All patients exhibiting symptoms were referred for the initiation of specialized medical care.

Following the identification of the SCA7 mutation in the discovery cohort, we implemented a diagnostic protocol in collaboration with the regional health authorities to voluntarily diagnose individuals with symptoms associated with SCA7, following a medical evaluation by a neurologist. We identified 105 additional individuals presenting clinical features of ataxia and visual dysfunction (decrease in visual acuity and partial or total color blindness) who voluntarily enrolled in the Symptomatic protocol between August 2015 and the present date. SCA7 mutations were confirmed in 91 of these individuals (Table 1). Of note, the SCA2 mutation was identified in the remaining 14 subjects from the central region of Veracruz.

A genealogical analysis revealed the existence of at least 391 asymptomatic individuals at a 50% risk for SCA7. In light of these findings, we took the decision to implement a pre-symptomatic diagnosis. Following the dissemination of the relevant information to each family at risk by the medical team responsible for the communities, 57 individuals at risk expressed an interest in being included in the pre-symptomatic diagnosis program. The program was initiated in September 2019 (Table 1). Nevertheless, only seven individuals enrolled in the testing program, and it is worth to note that only four participants (57%) completed the protocol, resulting in one patient being SCA7-positive and three negative. Consequently, the dropout rate was higher for individuals enrolled in the presymptomatic program than those in the symptomatic one. The sole SCA7 patient who completed the protocol cited multiple reasons for undergoing the test, including preparing mentally and physically to manage the disease, assessing the risk for their descendants, planning for the future, alleviating uncertainty, and informing and preparing their families regarding the disease. Notably, despite initial depressive symptoms, the participant’s response one year after diagnosis was positive, underscoring the importance of psychological support in navigating the result and fostering a constructive attitude toward supporting patients in the community.

Overall, 172 individuals were identified as carriers of the SCA7 mutation, of whom 141 exhibited symptomatic disease (classified as 107 AO and 34 EO) and 31 were identified as presymptomatic carriers. A total of 297 individuals identifying as being a relative at risk were recruited to participate in the SCA7 testing protocol, as illustrated in Figure 1.

In alignment with these findings, we estimated the prevalence of SCA7 in communities and municipalities of the central region of Veracruz State and other regions of Mexico where SCA7 patients were identified (Figure 2 and Table 1). Of particular interest, the prevalence in the community of Tlaltetela was found to be 1301.44 per 100,000 inhabitants, while in Tuzamapan, it was 337.61 per 100,000 inhabitants. This represents an increase from 59.26% to 109.43% of cases in these communities over a period of ~11 years in accordance with our previous report. Given that several communities have fewer than 6000 inhabitants, we decided to report the prevalence by municipality (populations exceeding 10,000 people). The data indicate that the municipalities of Tlaltetela, Cosautlán, and Coatepec presented the highest prevalence rates, with 448.87, 123.71, and 37.27 cases per 100,000 inhabitants, respectively (Table 1). The map illustrates the locations with the highest prevalence rates in the central region of Veracruz State, as well as the presence of isolated cases in other regions of the country.

### 2.2. Socioeconomic Status of SCA7 Patients in Veracruz State

The communities under examination are situated in regions of restricted accessibility, predominantly in the foothills of the Sierra Madre Oriental, in the Sierra de Chiconquiaco mountain range, on extrusive igneous rocks from the Quaternary, and basalt hills. As a consequence, the primary economic activities in this region are husbandry and agriculture, which include pine forests, fruit crops, wheat, sugar cane, and extensive coffee plantations. In the cohort of patients aged 12 years and older, 38.1% were illiterate, 22.6% had incomplete primary education, 23.9% had completed elementary school, 10.5% had completed junior high school, and only 4.9% had attained higher education. The data in question exhibit a notable discrepancy when compared to estimates for the state of Veracruz, where the illiteracy rate stands at 14.9%. Conversely, 50.2% of the population has completed primary education, 28.2% has completed elementary school, and 16.5% has finished high school. This is indicative of a relatively low academic level in the region, as measured against the standards set by the Organization for Economic Cooperation and Development (OECD).

It was determined that 70.3% of the population between the ages of 15 and 64 were employed. Most of these individuals were engaged in agricultural work, while many others were involved in other commercial activities. Conversely, 29.7% of the population in this age group were unemployed and lacked a regular source of income.

It is regrettable that social reintegration adaptation is constrained in these patients. Of the sample, 61% were male, with a mean age of 39 years (ranging from 10 to 74 years). All were of a productive age, yet afflicted with significant physical disabilities, with the majority exhibiting visual impairments that constrained their work capabilities. The mean monthly income of households with at least one affected individual was estimated to be USD 176.6, which is considerably below the average monthly salary of the population in the state of Veracruz, which is approximated at USD 326.20. Conversely, the costs associated with the medical care of the affected family members were approximately USD 48.54 per month, which also indicates a low socioeconomic level.

### 2.3. Origin of Mexican Patients of SCA7

In a previous study, the genomic SCA7-linked markers 3145G/A, D3S1287, D3S1228, and D3S3635 were used to evidence that all SCA7 Mexican patients shared the same haplotype, A-254-82-98 [7]. The aforementioned multi-loci combination is notably uncommon in both healthy relatives and the general Mexican population, indicating that a single ancestral mutation is the underlying cause of all SCA7 cases identified in the analyzed population. However, as a national reference center, we found nine new patients unrelated to those previously identified by using the SCA7-linked markers. It was unexpected that SCA7 patients from Monterrey/Torreón, Guerrero, and Guanajuato exhibited A-254-84-102, A-240-80-110, and A-254-84-106 haplotypes, respectively, which implies a different origin of the SCA7 mutation between patients from different regions of the country.

### 2.4. Genotype–Phenotype Correlations and Intergenerational Transmission

As expected, the results demonstrated that patients with the highest number of CAG repeats exhibited disease symptoms at an earlier age, presenting a phenomenon known as “anticipation”. A total of 42 distinct expanded alleles were identified, with lengths ranging from 34 to 138 CAG repeats (see Appendix A). It is noteworthy that within the AO phenotype, there is a subset of patients who develop the pathology only with mild symptoms after the fifth decade of life. In some cases, these patients may remain undiagnosed for an extended period due to the clinical manifestations being similar to those observed in aged individuals. Consequently, the expansion of CAG repeats was subjected to analysis in this subset of the AO cohort, revealing a range between 36 and 41 CAG repeats (mean = 38.77 CAG repeats). Then, they were assigned to the late-onset (LO) phenotype, as illustrated in Figure 3A. Patients who exhibited evident symptoms within the third to fifth decade of life were classified as having the classic SCA7 onset (CO), as illustrated in Figure 3A. The mean number of CAG repeats for this group of patients was 45.19, with a range of 39 to 53 repeats. In contrast, patients with early-onset (EO) phenotypes exhibited alleles between 43 and 138 CAG repeats (mean = 64.3 CAG repeats). A correlation between the age of onset and the number of CAG repeats revealed differential behavior between the three phenotypic groups (Figure 3B). In conclusion, the comprehensive characterization of a substantial number of SCA7 patients enabled the identification of distinct clinical subgroups.

Regarding the intergenerational transmission of expanded CAG repeats, our findings substantiate the conclusions previously reported [6]. The affected offspring of male parents exhibited larger CAG repeat expansions (mean CAG repeats = 12.9; SD = 17.05), which suggests greater instability. The occurrence of repeat expansions exceeding 10 CAGs was more prevalent through paternal transmission, as illustrated in Figure 4. In contrast, in the offspring of affected mothers, the expansions were predominantly small and discrete, indicating greater stability from one generation to the next (mean = 5.21 CAG repeats; SD = 4.79).

The final step was to categorize patients with SCA7 using a combination of clinical scales. Based on SARA (Scale for the Assessment and Rating of Ataxia) and INAS (Inventory of Non-Ataxia Signs), we conducted a comprehensive analysis of all SCA7 patients, subsequently categorizing them into three distinct clinical phenotypes (EO, AO, and LO). It is noteworthy that a positive correlation was observed between the SARA and INAS scales and the disease time span in SCA7 patients, specifically in the EO and AO phenotypes, with the sole exception of the LO phenotype. However, no correlation was found between age at onset and the aforementioned scales (Table 2). In contrast with the anticipated outcome, no evident correlation was found between any of the neurological scales and the size of the CAG repeat expansion.

#### Survival Analysis

A total of 36 deceased patients between 2013 and 2024 were registered during this research. The mean age at death was 46.35 years (SD: 27.15, ranging from 3 to 96 years). A significant inverse correlation was identified between the age of death and the number of CAG repeats presenting the expanded allele (r = −0.965, *p* < 0.01) (Figure 5A). Additionally, a significant positive correlation was observed between the age of death and the onset of symptoms (r = 0.965, *p* < 0.01). As evidenced by the phenotype of the disease, SCA7-EO patients have a limited life expectancy. Of the 17 identified deaths, the average age at death was 21.76 years (SD: 12.26, ranging from 3 to 44 years old). This is statistically significantly different from the average age at death (60.33 years old) found in patients with a CO phenotype (SD: 8.82, ranging from 46 to 72 years old) (see Figure 5A). Consistently, patients with an LO phenotype had an average age at death of 84 years (SD: 9.50, ranging from 73 to 96 years old), indicating a significantly longer life expectancy. However, no statistically significant differences were observed concerning SCA7-CO (Figure 5A). Due to the variability in the presentation of the pathology, it was observed that the disease-free survival rate for patients with EO was markedly inferior to that of patients with CO or LO (Logrank test, w2 1/4 62, *p* < 0.0001), with the disease-free survival rate of LO patients even lower than that of CO patients (Logrank test, w2 1/4 5.7, P 1/4 0.017) (Figure 5B). Undoubtedly, the deaths of patients with EO and AO phenotypes were associated with complications related to the pathology, while LO patients were linked to age-related intrinsic causes. However, specific data on the cause of death are lacking due to the absence of forensic doctors in the communities. In conclusion, it is evident that survival depends on the age at which the pathology manifests. 

## 3. Discussion

This study provides an update to a longitudinal long-term study of SCA7 in the Mexican population. A comprehensive analysis of an unusually large cohort of Mexican SCA7 patients was conducted for ~11 years (since 2013–2014), covering different areas of research, including molecular diagnosis, haplotype analysis to determine the SCA7 mutation origin in our country, medical care including neurological assessment, genetic counselling, physical rehabilitation, psychological management, and social issues. Furthermore, the continuity of this study allowed us to obtain genotype–phenotype correlations, and to determine the influence of parental/maternal transmission of mutant alleles on CAG repeat stability. Notably, the haplotype analysis of patients recruited from regions other than Veracruz State revealed the possibility of having more than one origin of SCA7 in Mexico. 

The frequency of SCA7 is estimated to be <1/100,000 worldwide, yet the southeast region of Mexico exhibits the highest concentration of cases described in a specific geographic region [11]. In this locality of only 1200 km^2^, 172 carriers of the SCA7 mutation were identified. The worrisome increase in SCA7 prevalence in this region can be attributed, at least in part, to its geographical location with limited accessibility and the low socioeconomic status of the population. In this context, our research group, comprising, medical geneticists, neurologists, and rehabilitation physicians, developed a series of strategies for the diagnosis and management of patients, with the objective of improving the quality of life of patients and their families. The initial phase of the project involved the implementation of clinical and molecular diagnoses of the disease. It was essential to train healthcare professionals on how to diagnose the disease and to disseminate information to the general population regarding its transmission and fundamental clinical characteristics. This required promoting a culture of understanding the disease while ensuring that patients and their families were not subjected to further distress.

It is worth noting that the increased prevalence of the disease results in a higher number of at-risk individuals of reproductive age, which would ultimately lead to an increase in the number of SCA7 in subsequent generations. As an attempt to ameliorate this potential consequence, a predictive diagnosis program (genetic counseling and clinical examination) was established. The implementation of preventive diagnostics has demonstrated efficacy in the management of various polyglutamine diseases, including Huntington’s disease in Mexico [27,28,29,30] and SCA2 and SCA3 in Cuba and Brazil, respectively [31,32,33,34,35]. Nonetheless, the protocol was not more widely accepted by Veracruz communities. This may be attributed to the fact that the predictive testing was conducted at a medical center situated over 40 km away from the communities. Consequently, patients lacked the financial resources to transport themselves to the clinic and take their physical therapy and counseling sessions. To solve this problem, it would be necessary to provide such services within the communities. In addition, most of the population was reluctant to undergo genetic testing, alluding to concerns about their ability to cope with adverse results and social stigmatization. In contrast, other studies have shown that most individuals find predictive testing to be beneficial regardless of the result [36,37,38,39]. Therefore, it is evident that this program necessitates a considerable period of adjustment for the Mexican population, as well as the input of professionals to address the psychological implications [40].

Although the clinical presentation of SCA7 is influenced by genetic and extrinsic modifying factors, it is mainly determined by the length of the CAG repeats tract. Patients with early-onset (EO) phenotypes have CAG expansions between 43 and 138 repeats, and exhibit visual impairment, movement disorders, spasticity, and autonomic abnormalities, while patients with adult-onset (AO) phenotypes have an average of 51.30 ± 7 CAG repeats [6] and moreover exhibit the classic manifestations of SCA7, including cerebellar features and pyramidal signs [5]. Interestingly, the analysis of our large cohort enabled the identification of a new SCA7 phenotype, designated as late-onset (LO). This group of patients barely exceeded the pathological threshold of CAG repeats with an average of 51.30 ± 7.38 and exhibited very mild clinical manifestations after the sixth decade of life. As anticipated, the phenotype observed in patients ultimately determined their life expectancy. During the study, 36 patients unfortunately succumbed to their illnesses, and further correlation with their phenotype demonstrated that those with the EO phenotype had a reduced life expectancy, which is consistent with previous reports [2,12]. These patients rapidly lost their visual capacity following the onset of the pathology, requiring specialized care that included psychological support from primary caregivers and family members.

Regarding patients exhibiting the CO phenotype, it was found that they had a longer life expectancy, thereby making it more feasible to provide them with medical care, including physical rehabilitation, which would improve their quality of life [26]. Finally, patients with the LO phenotype exhibited the highest life expectancy, and they could virtually lead a normal life, although they are more likely to die from common age-related diseases and therefore present a lower survival than patients with a CO phenotype. A variety of clinical scales have been developed with the objective of semiquantitatively assessing cerebellar and non-cerebellar characteristics in patients with different SCA disorders [41,42]. These tools have facilitated the monitoring of the natural progression of these diseases and the evaluation of the efficacy of clinical interventions. It is evident that neurological scales correlate with disease progression across all phenotypes, with the exception of the LO phenotype. This is due to the fact that the LO phenotype presents with very mild signs and symptoms, which may go unnoticed by the patient and therefore may not be included in the assessment. Furthermore, the threshold for onset was comparable to that observed in individuals of advanced age. Conversely, the age of onset and the size of the CAG repeat expansion demonstrated an inverse and direct proportional relationship, respectively. However, the extensive population allowed for the observation of high variability in disease presentation and repeat numbers, which precluded the identification of statistically significant correlations. This indicated the potential influence of external factors in addition to the possibility of modifier genes, which may have contributed to the observed inconsistency in subjective scale assessments across different phenotypes. It is therefore vital to employ objective markers for the evaluation of the pathology.

The last finding of our study that merits discussion was the origin of the pathology in Mexico. Our previous studies employing genetic markers linked to the *ATXN7* gene indicated that the disease was initiated in Mexico through a single mutation via a founder effect [7]. However, the observation that other patients recruited from outside the region possessed a distinct haplotype from the Veracruz patient cohort implied that the pathology may have originated in Mexico from at least two distinct sources. The findings and their implications should be discussed in the broadest context possible. Future research directions may also be highlighted.

## 4. Materials and Methods

### 4.1. Participants

The cohort under study comprised 297 subjects, which comprised 172 SCA7 carriers diagnosed and their relatives at risk of developing the condition, who were screened for the SCA7 mutation by molecular testing. The participants were enrolled by the National Rehabilitation Institute Luis Guillermo Ibarra Ibarra (INR-LGII) and the Rehabilitation and Social Inclusion Center of Veracruz (CRISVER) from the central region of Veracruz State, Mexico. Patients with secondary ataxias resulting from alcoholism, vascular pathology, malignant neoplasms, congenital anomalies, neuropharmacological treatment, autoimmune or inflammatory diseases, and other non-genetic causes were excluded from the study. All subjects provided signed informed consent as volunteers. The study was approved by the INRLGII Ethics/Research Committee (protocol INRLGII-15/12). All procedures conducted in this study were performed in accordance with the ethical standards set forth in the Helsinki Declaration. Finally, a structured questionnaire was administered to each participant to collect sociodemographic data, including educational level, occupation, income, and expenses related to their condition. These data were then compared with state-level information provided by the National Institute of Statistics and Geography (INEGI). 

### 4.2. Testing Programs to Detect SCA7 Mutation

Three protocols were established for the detection of the SCA7 mutation in the studied communities (Figure 1). In the initial phase, a protocol was implemented to identify the causative mutation in the founder population in the central region of Veracruz State. Blood samples were collected from voluntary SCA patients and their relatives in their communities for research purposes. Subsequently, we established a symptomatic diagnosis with the objective of providing specialized and multidisciplinary medical care to patients with SCA7, with the aim of reducing the negative social and emotional impact caused by the disease. During this stage, we analyzed samples from different regions of Mexico. Thereafter, a multidisciplinary approach was undertaken, incorporating the disciplines of genetics, clinical psychology, ophthalmology, phoniatrics, and comprehensive rehabilitation management, as well as palliative care. This approach included the analysis of genetic factors, such as medical history, pedigree analysis, and counselling, as well as clinical follow-up, which entailed examination by the Scale for the Assessment and Rating of Ataxia (SARA) and the Inventory of Non-Ataxia Signs (INAS). Additionally, psychological assessment was conducted to identify any indications of anxiety or depression related to the patient’s SCA7 history or other influencing factors. This was followed by the implementation of appropriate therapy and constant re-evaluation. Rehabilitation management was also provided through occupational and physical therapy, speech therapy, and hydrotherapy.

In the final stage, the social, ethical, and legal conditions necessary to provide a presymptomatic diagnosis were established. This protocol included an initial neurological examination and two genetic counseling sessions, as well as at least three psychological screening evaluations prior to genetic testing. The objective of these evaluations was to ascertain the applicants’ psychological capacity to receive the genetic test results. At the time of disclosure, participants were informed of their potential carrier status, which was also relevant for at-risk relatives. Then, psychological follow-up evaluations were conducted at one-week, four-week, six-month, and one-year intervals following the disclosure of the genetic testing results. This protocol was carried out at the facilities of the Reference Center for the care of patients with SCA7 in Veracruz State CRISVER, approximately 40 km from the communities.

Finally, an investigation was conducted into the sociodemographic data, psychological background, and past medical history of all individuals who had requested inclusion in the testing programs, based on an analysis of the individual interviews and medical records belonging to each individual. During the enrollment process, several matters of concern were addressed, including motivations for requesting the genetic test, the predicted implications according to test results, the intended recipients of the results, and the purpose of such a disclosure.

### 4.3. Molecular Diagnosis of SCA7

The analysis of CAG repeats was conducted in the laboratory of Genomic Medicine in INR-LGII, employing fluorescent PCR and capillary electrophoresis. In brief, genomic DNA was extracted from peripheral blood leukocytes using a Gentra Puregene blood kit (Qiagen, Hilden, Germany). Subsequently, fluorescent multiplex PCR was conducted on a Veriti thermal cycler (Applied Biosystems, Foster City, CA, USA) using the DNA template (60 ng) and the chimeric primers described by Dorschner et. al. [43]. Finally, the PCR-amplified products were subjected to capillary electrophoresis on an ABI PRISM 3730XL Genetic Analyzer (Applied Biosystems, Foster City, CA, USA) [7,11]. The allelic distribution of the SCA7 mutation was analyzed (Appendix A). Subjects presenting a single allele when genotyped for SCA2 and SCA7 were re-genotyped by triplet repeat primed (TP)-PCR to eliminate the possibility of false negatives resulting from the presence of pathological large alleles (>100 CAG repeats), as previously described by Cagnoli et al. [44].

### 4.4. Specific Neurological Assessments

All subjects were interviewed to obtain their clinical history and were clinically assessed with standard neurological examinations following the Mayo Clinic procedure [6]. Motor control of the limbs was evaluated to identify dysmetria and dysdiadochokinesia, while cerebellar signs were explored to detect gait ataxia, tremor, or dysarthria. An examination of the cranial nerve was conducted to determine any alterations in ocular movement, while an exploration of extrapyramidal signs identified any involuntary movements. Additionally, muscle strength and pathological reflexes were assessed. The SARA was employed for the evaluation of ataxia-associated symptoms [45]. At the same time, we sought to identify extracerebellar features through the use of the INAS [46]. In accordance with our previous report, the SCA7 patients were initially classified into two main phenotypes: The patients were classified as AO and EO based on the age at which the first symptoms were reported, which was found to be above or below 20 years of age, respectively [6].

### 4.5. Haplotype Analysis

We conducted a haplotype analysis as previously described [7]. Four markers linked to the SCA7 gene were utilized: the intragenic SNP 3145G/A and three centromeric microsatellite markers (D3S1228, D3S1287, and D3S3635), which span the 5.21-cM region of chromosome 3p harboring the ATXN7 gene. The intragenic SNP was analyzed by a real-time PCR assay on a StepOne™ Thermal Cycler (Applied Biosystems, Foster City, CA, USA). Similarly, the microsatellite markers were amplified by PCR and then analyzed on a DNA analyzer 3730xl sequencer (Applied Biosystems, Foster City, CA, USA). To ascertain the size of each allele, we employed the GeneScan™ 500 LIZ Size Standard (Applied Biosystems, Foster City, CA, USA). Alleles were designated subjectively based on their size in base pairs (bps).

### 4.6. Statistical Analysis

Descriptive statistics were used to depict the central tendencies and dispersion of the variables under study. To test the normality of data, the Shapiro–Wilk tests was performed. The prevalence data were calculated by this equation: [(Number of cases) (100,000 inhabitants)]/total population of inhabitants for each community and/or municipality. A regression analysis was conducted to ascertain the relationship between the age of onset, CAG number of repeats, and age of death. In conclusion, the cumulative probability of developing SCA7 at a specific age, given a particular CAG repeat expansion, was calculated for nearly all SCA7 phenotypes. All calculations were performed using the statistical software package SPSS (version 26.0).

## 5. Conclusions

In summary, although SCA7 is regarded as a rare disease worldwide, it displays a relatively high prevalence in Mexico, in a limited area of central Veracruz. The implementation of a comprehensive care program for this population, which included medical care, genetic testing, and rehabilitation, as well as the longitudinal follow-up of patients, resulted in an improvement in patient living conditions and a generation of results that facilitated our understanding of the disease. The major findings of our study include the influence of paternal transmission on CAG repeat instability, the correlations between the length of CAG repeats and clinical outcomes and between the disease phenotype and life expectancy, the identification of a subgroup of patients with a late-onset phenotype, and the evidence of more than one origin of SCA7 in the Mexican population. The continuation of this study will facilitate the discovery of new clinical and molecular characteristics of SCA7, the identification of biomarkers and potential therapeutic targets, and the design of effective rehabilitative physical therapies.

## Figures and Tables

**Figure 1 ijms-25-10750-f001:**
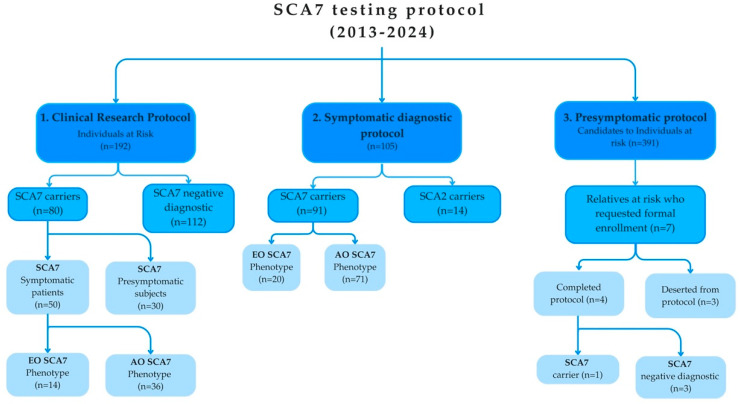
Flowchart showing the outcome of the SCA7 testing program in Mexico over the course of an 11-year period.

**Figure 2 ijms-25-10750-f002:**
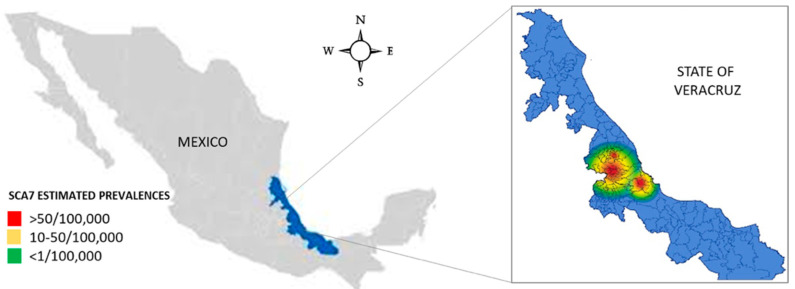
Heatmap illustrating the Geographic Prevalence of Mexican Patients with SCA7. These municipalities and communities encompass a small region of 1200 km^2^.

**Figure 3 ijms-25-10750-f003:**
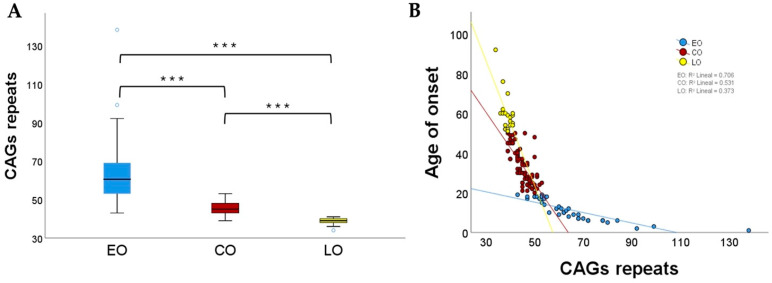
Relationship between phenotypes and CAG expansion. (**A**) Size of CAG repeats in each clinical phenotype. Early Onset (EO), Classic Onset (CO) and Late Onset (LO); significant differences between groups were observed (*** *p* < 0.001), value obtained by Kruskal–Wallis test. (**B**) Correlation between age at onset and the number of CAG repeats in SCA7.

**Figure 4 ijms-25-10750-f004:**
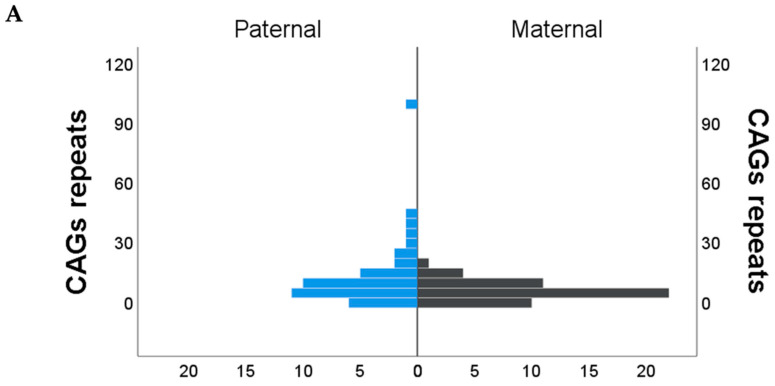
Intergenerational instability of the SCA7 CAG repeats. The illustration depicts the frequency of expansion, contraction, or no change in the transmission of SCA7 alleles between male and female carriers. (**A**) Parental transmission frequency; significant differences between groups were observed *p* = 0.003, value obtained using Mann Whitney *U* test. (**B**) Frequency of intergenerational expansion. The proportions of expansion ranges, defined as the number of repeats, are illustrated. A significant difference between groups was observed; *p* = 0.014, value obtained using Fisher’s Exact test.

**Figure 5 ijms-25-10750-f005:**
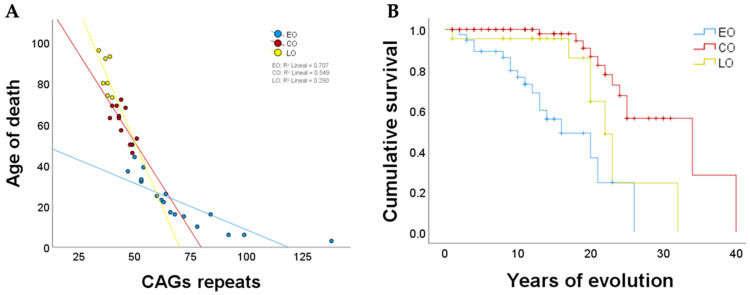
Survival analysis of patients with SCA7. (**A**) Correlation between age at onset and the number of CAG repeats in SCA7; (**B**) Kaplan–Meier survival analysis on EO, CO, and LO SCA7 patients.

**Table 1 ijms-25-10750-t001:** Prevalence for patients with SCA7 in the central region of the state of Veracruz.

Municipality	Inhabitants	SCA7 Patients	Prevalence Rate
Tlaltetela	16,486	74	448.87
Cosautlan	16,167	20	123.71
Coatepec	93,911	35	37.27
Teocelo	16,957	6	35.38
Naolinco	22,835	4	17.52
Acajete	9701	1	10.31
Tezonapa	54,537	2	3.67
Cuitlahuac	28,875	1	3.46
Xalapa	488,531	14	2.87
Cordoba	204,721	1	0.49
Emiliano Zapata	85,489	1	1.17

**Table 2 ijms-25-10750-t002:** Correlation analysis of the SARA and INAS scores with clinical and genetic parameters.

	SARA		INAS	
** *SCA7 patients* **	** *r* **	** *p* **	** *r* **	** *p* **
Disease duration	**0.515**	**0.001**	**0.512**	**0.001**
Age at onset	−0.151	0.172	−0.009	0.954
CAG Repeats	0.139	0.21	−0.011	0.944
** *EO phenotype* **				
Disease duration	**0.717**	**0.001**	**0.425**	**0.049**
Age at onset	0.299	0.122	0.265	0.305
CAG Repeats	−0.246	0.24	−0.345	0.191
** *AO phenotype* **				
Disease duration	0.751	0.001	0.631	0.001
Age at onset	−0.047	0.761	0.008	0.97
CAG Repeats	**0.347**	**0.021**	0.181	0.409
** *LO phenotype* **				
Disease duration	0.168	0.622	0.572	0.313
Age at onset	0.049	0.886	0.326	0.592
CAG Repeats	0.23	0.485	0.147	0.813

Significant values (*p* < 0.05) are indicated in bold.

## Data Availability

Not Applicable.

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
