# Peer review of "Current Overview of Spinocerebellar Ataxia Type 7 in Mexican Population: Challenges in Specialized Care for a Rare Disease"

_ijms, 2024, doi:10.3390/ijms251910750_

Round 1

Reviewer 1 Report

Comments and Suggestions for Authors

In this original contribution entitled “Current Overview of Spinocerebellar Ataxia Type 7 in Mexican population: Challenges in specialized care for a rare disease”, César M. Cerecedo-Zapata, Yessica S. Tapia-Guerrero et al. described the result of a program for the diagnosis, medical care, and long-term follow-up, covering from 2013 to 2024, of patients affected by SCA7 in Veracruz State (Mexico). 

The manuscript is well structured, English is well written. The conclusions are sufficient, partially concrete. Authors highlight the importante of their protocol, which includes medical care, genetic testing, and rehabilitation, as well as the longitudinal follow-up of patients. However, not all the informations are fully explained or presented. Thus, I have major and some minor concerns.

Major concerns:
- It my opinion, it is difficult to understand the real difference of the “clinical research protocol” and “symptomatic testing”, and overall, your flowchart. Even if the protocol is described in the methods section, which came after the result description, I would suggest to briefly present it and the flowchart in the result paragraph.

- Same as before: the concept of individuals “being at risk” should be presented better to make all the reading more understandable

- Figure 1: the sentence “*SCA7 testing positive in 100%” is not understandable. Can you comment on that?

- In the paragraph “Socioeconomic status of SCA7 patients in Veracruz State” authors described some information regarding the socioeconomic status of their cohort. 
1) How did the take this information? In the methods section a description of the process would be needed;
2) It would be interesting to put in context the data from the patient’s cohort with the ones from the general population in Velacruz State.

- I would add more information the paragraph “Genotype-Phenotype Correlations and intergenerational transmission” which is central in your paper. 
1) You defined LO patients having “mild symptoms”; can you present some data regarding SARA score (severity) correlation with age of onset? (you mentioned SARA in the methods part);
2) Can you please provide more information regarding the phenotype (neurological and non-neurological)? In the methods section entitled “specific neurological assessments” you stated that you collected phenotypic information through clinical examination and scale (SARA, ICANS). It would be interesting to present those data you collected (in a table for example), correlating them with CAG repeats and age of onset for instance. I believe this could add more relevance to your work.

- In the “survival analysis” paragraph it would be interesting to state the cause of death of the patients (if data are available).

Minor concerns:
- line 76, the sentence “In this context, it is noteworthy that we found” it’s not proper English 
- line 96, I would change the term “commencement”
- - line 98-99, please define the meaning of “visual disturbance” 
- line 188-189, the sentence “the expansion of CAG repeats was subjected to analysis in this cohort of patients” does not follow the rationale of your methods. LO is not a different cohort of patients but I would call it a subset of AO cohort.
- line 207, I would change “mean = 12.9, SD = 17.05, p=0.003” with “mean CAG repeats= 12.9, SD = 17.05, p=0.003”
- line 324-327, eliminate the sentence “Authors should discuss the results and how they can be interpreted from the perspective of previous studies…”
- Figure 1: SCA2 number should be 14
- in the paragraph “Genotype-Phenotype Correlations and intergenerational transmission”  mean number of CAG repeats for LO group of patients is not presented

Comments on the Quality of English Language

English is well written, manuscript easy to read. Minor revisions are needed.

Author Response

REVIEWER 1

In this original contribution entitled “Current Overview of Spinocerebellar Ataxia Type 7 in Mexican Population: Challenges in specialized care for a rare disease”, César M. Cerecedo-Zapata, Yessica S. Tapia-Guerrero et al. described the result of a program for the diagnosis, medical care, and long-term follow-up, covering from 2013 to 2024 of patients affected by SCA7 in Veracruz State (Mexico). 

The manuscript is well structured, English is well written. The conclusions are sufficient, partially concrete. Authors highlight the importance of their protocol, which includes medical care, genetic testing, and rehabilitation, as well as the longitudinal follow-up of patients. However, not all the information is fully explained or presented. Thus, I have major and some minor concerns.

Major concerns:

  1. It my opinion, it is difficult to understand the real difference of the “clinical research protocol” and “symptomatic testing”, and overall, your flowchart. Even if the protocol is described in the methods section, which came after the result description, I would suggest to briefly present it and the flowchart in the result paragraph.
  • In response to the reviewer's constructive criticism, we have revised the Manuscript to provide a more comprehensive explanation of the diagnostic protocol used (see Results, first paragraph). We have highlighted the objectives of each diagnostic phase and the rationale for their implementation. Additionally, we have modified the presentation of Figure 1 to enhance its clarity and facilitate comprehension.

  • Same as before: the concept of individuals “being at risk” should be presented better to make all the reading more understandable
  •  
  • In accordance with this pertinent observation, "the relative risk" is now defined in the first paragraph of the "Results" section.   

  1. Figure 1: the sentence “*SCA7 testing positive in 100%” is not understandable. Can you comment on that?
  • In concordance with this observation, Figure 1 was modified and the referred statement was changed properly.

  1. In the paragraph “Socioeconomic status of SCA7 patients in Veracruz State” authors described some information regarding the socioeconomic status of their cohort. 
    How did they take this information? In the methods section a description of the process would be needed;
  • We now clarified the procedure that was carried out to collect the socioeconomic data of patients. See Material and Methods, Participants section.

  1. It would be interesting to put in context the data from the patient’s cohort with the ones from the State of Veracruz.
  • We now provide an overview of the current situation of patients and their families in relation to the state of Veracruz, as indicated by the latest federal census. See Results, 2.2 Socioeconomic status of SCA7 patients in Veracruz State.

  1. I would add more information the paragraph “Genotype-Phenotype Correlations and intergenerational transmission” which is central in your paper. 
    1) You defined LO patients having “mild symptoms”; can you present some data regarding SARA score (severity) correlation with age of onset? (you mentioned SARA in the methods part);
  • Following this opportune observation. Correlation of clinical scales (SARA and INAS) values with genetic and demographic characteristics is now presented in new Table 2. These results are now better explained and discussed.

  1. Can you please provide more information regarding the phenotype (neurological and non-neurological)? In the methods section entitled “specific neurological assessments” you stated that you collected phenotypic information through clinical examination and scale (SARA, ICANS). It would be interesting to present those data you collected (in a table for example), correlating them with CAG repeats and age of onset for instance. I believe this could add more relevance to your work.
  • In line with this, we present a new Table 2 that shows a correlation between the neurological scales and either the length of the CAG repeats tract, age at onset of the disease, and disease duration. These data are presented and discussed in the “Results” and “Discussion” sections.

  1. In the “survival analysis” paragraph it would be interesting to state the cause of death of the patients (if data are available).
  • A brief explanation has been added regarding the status of deceased patients and the impossibility of determining the cause of death due to the lack of medical examiners in the communities.

Minor concerns:
1. line 76, the sentence “In this context, it is noteworthy that we found” it’s not proper English

  • We have rephrased this sentence to add clarity.

  1. line 96, I would change the term “commencement”
  • We changed the term as suggested by the reviewer.

  1. line 98-99, please define the meaning of “visual disturbance” 

  • We have now defined the meaning of visual dysfunction for a better understanding of the term.

  1. line 188-189, the sentence “the expansion of CAG repeats was subjected to analysis in this cohort of patients” does not follow the rationale of your methods. LO is not a different cohort of patients but I would call it a subset of AO cohort.
  • In accordance with this appropriate comment, we have clarified in the text that the LO phenotype is a subset of the AO cohort.
  1. line 207, I would change “mean = 12.9, SD = 17.05, p=0.003” with “mean CAG repeats= 12.9, SD = 17.05, p=0.003”
  • The suggested change was made.

  1. line 324-327, eliminate the sentence “Authors should discuss the results and how they can be interpreted from the perspective of previous studies…”
  • We removed the sentence in accordance with the reviewer's suggestion.
  1. Figure 1: SCA2 number should be 14

- The error in Figure 1 was mended.    

  1. in the paragraph “Genotype-Phenotype Correlations and intergenerational transmission” mean number of CAG repeats for LO group of patients is not presented
  • The mean number of CAG repeats for LO group is now shown in the paragraph of Genotype-Phenotype correlations.

Reviewer 2 Report

Comments and Suggestions for Authors

The manuscript "Current Overview of Spinocerebellar Ataxia Type 7 in Mexican population: Challenges in specialized care for a rare disease" describes the current status and new findings (clinical, genetic, demographic and epidemiological aspects) of SCA7 in Mexican population, aiming to contribute to the diagnosis, clinical management and potential treatment of the rare SCA7 disorder in Mexican cohort. SCA7 is an autosomal dominant cerebellar ataxia type II that is characterized by progressive ataxia, motor system abnormalities, dysarthria, dysphagia and retinal degeneration leading to progressive blindness. The worldwide prevalence of the disease is estimated to be less than 1/100,000 and it is thought to account for about 2-4 % of all forms of spinocerebellar ataxia. However, higher prevalence is reported in some populations such as in Scandinavia or South Africa. The submitted manuscript abundantly gives an overview of SCA7 in Mexican population which will aid in the evidence based management of this rare disorder.

After going through the manuscript, I have following comments for the authors.

1.     Is the prevalence of SCA7 in Mexico and other south American countries known?

2.     Differential diagnoses are known to include lipid storage diseases (such as neuronal ceroid lipofuscinosis) and Leber hereditary optic neuropathy (LHON). Were these aspects reported in some included patients?

Comments on the Quality of English Language

Minor grammatical corrections and syntax adjustments recommended.

Author Response

REVIEWER 2

The manuscript "Current Overview of Spinocerebellar Ataxia Type 7 in Mexican population: Challenges in specialized care for a rare disease" describes the current status and new findings (clinical, genetic, demographic and epidemiological aspects) of SCA7 in Mexican population, aiming to contribute to the diagnosis, clinical management and potential treatment of the rare SCA7 disorder in Mexican cohort. SCA7 is an autosomal dominant cerebellar ataxia type II that is characterized by progressive ataxia, motor system abnormalities, dysarthria, dysphagia and retinal degeneration leading to progressive blindness. The worldwide prevalence of the disease is estimated to be less than 1/100,000 and it is thought to account for about 2-4 % of all forms of spinocerebellar ataxia. However, higher prevalence is reported in some populations such as in Scandinavia or South Africa. The submitted manuscript abundantly gives an overview of SCA7 in Mexican population which will aid in the evidence based management of this rare disorder.

After going through the manuscript, I have following comments for the authors.

1.Is the prevalence of SCA7 in Mexico and other south American countries known?

  • In response to this insightful comment, we have added information regarding the current prevalence of SCA7 in Mexico and Latin America, with their respective updated references. It is regrettable that there is a lack of data at the national level; instead, the available information is limited to certain communities and regions of a few Latin American countries. This further highlights the significance of the present study.

2.Differential diagnoses are known to include lipid storage diseases (such as neuronal ceroid lipofuscinosis) and Leber hereditary optic neuropathy (LHON). Were these aspects reported in some included patients?

  • It is unfortunate that the data in question were not found in any of the evaluated patients. It is noteworthy that the exclusion criteria only included cases of ataxia that were not linked to pathological effects. The SCA7-negative cases were verified as another type of spinocerebellar ataxia, specifically "SCA2," as stated in the text.

  1. Minor grammatical corrections and syntax adjustments recommended.

-     The text of the manuscript was reviewed in detail and some typographical errors found were mended.

Reviewer 3 Report

Comments and Suggestions for Authors

This study updates research on Spinocerebellar ataxia type 7 (SCA7) in a large Mexican cohort. The authors revealed that disease severity, determined by CAG repeat length, defines three phenotypes (early, classical, and late onset), with early-onset patients facing the most severe outcomes. Furthermore, paternal transmission of mutant alleles increases CAG repeat instability, and haplotype analysis indicates different genetic origins for patients outside Veracruz. This study provides insights for improved diagnosis, genetic counseling, rehabilitation, and therapeutic strategies.

A minor revision of the study is required before it can be published in IJMS:

  1. The axis texts in both Figures 3 and 5 are too small to be easily recognized.
  2. In line 39, “different genetic origin” should be corrected to “different genetic origins,” and “longitudinal observation” should be changed to “longitudinal observations.”
  3. In line 211, remove the extra word “small.”
  4. In line 212, “repats” should be corrected to “repeats.”
  5. I do not think t-tests are appropriate for proportions in Figures 4A and 4B. A z-test, chi-square test, or Fisher’s exact test would be more suitable.

Author Response

This study updates research on Spinocerebellar ataxia type 7 (SCA7) in a large Mexican cohort. The authors revealed that disease severity, determined by CAG repeat length, defines three phenotypes (early, classical, and late onset), with early-onset patients facing the most severe outcomes. Furthermore, paternal transmission of mutant alleles increases CAG repeat instability, and haplotype analysis indicates different genetic origins for patients outside Veracruz. This study provides insights for improved diagnosis, genetic counseling, rehabilitation, and therapeutic strategies.

A minor revision of the study is required before it can be published in IJMS:

  1. The axis texts in both Figures 3 and 5 are too small to be easily recognized.
  • The presentation of Figures 3 and 5 was improved accordingly with the observation of the reviewer.

  1. In line 39, “different genetic origin” should be corrected to “different genetic origins,” and “longitudinal observation” should be changed to “longitudinal observations.”

  • The mentioned sentences were corrected properly.

  1. In line 211, remove the extra word “small.”

  • The suggested change was done.

  1. In line 212, “repats” should be corrected to “repeats.”

  • This error was mended.

  1. I do not think t-tests are appropriate for proportions in Figures 4A and 4B. A z-test, chi-square test or Fisher’s exact test would be more suitable.

  • We appreciate the reviewer's comments and want to clarify the statistical approach used. For Figure 4A, we are not comparing proportions, but rather analyzing the changes in the CAG repeats, which are semicontinuous values. Since this data does not follow a normal distribution, as shown by the Shapiro-Wilk and Kolmogorov-Smirnov tests, we chose to use the non-parametric Kruskal-Wallis test to compare the medians between the groups, which is methodologically appropriate. Additionally, the results show a p-value of 0.003, indicating statistically significant differences in the distribution of changes in the CAG repeats between the paternal and maternal lines.

  • In Figure 4B, where we compare the frequencies of categorized expansions, we opted for Fisher’s exact test because some cells have fewer than 5 cases. Although we considered the chi-square test, it is not suitable when expected frequencies are so small, as the results could be unreliable under these conditions. Fisher’s exact test shows a p-value of 0.014, indicating statistically significant differences in the distribution of categorized expansions between the paternal and maternal lines.
  • We again appreciate the reviewer's suggestions.

Round 2

Reviewer 1 Report

Comments and Suggestions for Authors

The revision performed by the authors improved the overall quality of the manuscript and of the scientific presentation. I would consider the work ready for publication in IJMS.